# Dominant control of temperature on (sub-) tropical soil carbon turnover

Vera D. Meyer [1] ✉, Peter Köhler [2], Nadine T. Smit[1,3], Julius S. Lipp[1], Bingbing Wei[2], Gesine Mollenhauer [1,2,4] & Enno Schefuß [1] ✉

Carbon storage in soils is important in regulating atmospheric carbon dioxide ($CO_2$). However, the sensitivity of the soil-carbon turnover time ($\tau_{soil}$) to temperature and hydrology forcing is not fully understood. Here, we use radiocarbon dating of plant-derived lipids in conjunction with reconstructions of temperature and rainfall from an eastern Mediterranean sediment core receiving terrigenous material from the Nile River watershed to investigate $\tau_{soil}$ in subtropical and tropical areas during the last 18,000 years. We find that $\tau_{soil}$ was reduced by an order of magnitude over the last deglaciation and that temperature was the major driver of these changes while the impact of hydroclimate was relatively small. We conclude that increased $CO_2$ efflux from soils into the atmosphere constituted a positive feedback to global warming. However, simulated glacial-to-interglacial changes in a dynamic global vegetation model underestimate our data-based reconstructions of soil-carbon turnover times suggesting that this climate feedback is underestimated.

Globally, soils store more than twice as much carbon as the atmosphere[1,2]. Since the soil carbon cycle is sensitive to climate change and human activities[1,3,4], future warming, shifts in precipitation patterns and land use might perturb the soil-carbon storage and subsequently result in positive feedbacks on global warming via $CO_2$ release into the atmosphere[1,5]. Soil carbon storage is regulated by carbon influx (fixation through net primary production; NPP) and efflux. The latter is controlled by microbial respiration, soil erosion and fire emissions[2,5]. These processes determine $\tau_{soil}$ defined[6] as:

$$\tau_{soil} = \frac{C_{soil}}{f} \tag{1}$$

where $C_{soil}$ is the soil carbon stock (in kgC m$^{-2}$) and f either the carbon influx (NPP) or the efflux (in kgC m$^{-2}$ yr$^{-1}$). Under steady state conditions influx and efflux are equal[7]. Turnover times are critical components in carbon cycling for constraining the time scales of carbon exchange between different reservoirs. $\tau_{soil}$ depends on soil temperature[3,4,8] and moisture content[3,4] but also on chemical properties[9–11] and soil fertility[9,11]. Temperature effects on $\tau_{soil}$ are widely observed across the globe[4] while hydroclimate may exert strong control in low to mid latitudes where it may override temperature effects[4,12,13]. However, the key controls on $\tau_{soil}$ and their interactions are still debated[3,10,12]. This forms a major open question in tropical and subtropical regions where combined effects of future warming and precipitation changes may be amplified or attenuated depending on whether warming will be accompanied by drier or wetter conditions[12]. One compromising factor of understanding turnover times and their environmental controls is that our knowledge mostly relies on short-term observations of years to decades (e.g. ref. 12). The geological record is a unique and important means to gain information about centennial to millennial time scales. Characterized by global warming, hydroclimate change and rising atmospheric $CO_2$[14,15] the last deglaciation (~18,000–11,000 yrs before present (BP), henceforth referred to as 18–8 kyrs BP) is a promising analogue to investigate climate–soil-carbon turnover interactions over several millennia. Unfortunately, proxy data constraining deglacial changes in soil carbon storage and $\tau_{soil}$ in the tropics and subtropics are very scarce and existing data provide qualitative estimates only[16]. The aim of this study is to provide quantitative

[1]MARUM – Center for Marine Environmental Sciences, University of Bremen, Bremen, Germany. [2]Alfred-Wegener-Institut Helmholtz Zentrum für Polar- und Meeresforschung, Bremerhaven, Germany. [3]Present address: Bruker Daltonics GmbH & Co. KG., Bremen, Germany. [4]Present address: Department of Geosciences, University of Bremen, Bremen, Germany. ✉e-mail: vmeyer@marum-alumni.de; eschefuss@marum.de

glacial-to-Holocene reconstructions of $\tau_{soil}$ in the (sub-)tropics and to identify the major environmental controls.

We investigate how $\tau_{soil}$ changed in the Nile River catchment during the last 18 kyrs. With a length of 6650 km the Nile River is the longest river in the world. Spanning 35° of latitude (4 °S to 31 °N) in northeastern Africa and draining a catchment of nearly 3 million km² it extends over several vegetation zones (rainforest in the headwaters, savannah, Sahara Desert and the Mediterranean zone at the coast; Fig. 1). Mediterranean sediments supplied by the Nile River load form a powerful recorder of climate change integrating over this vast catchment area and thus being representative of the entire northern African tropics and subtropics. During the last deglaciation the northern African climate warmed[17,18] and humid conditions during the African Humid Period (AHP, 14.8–5.5 kyrs BP)[19] allowed for plants and permanent water bodies to persist in the nowadays barren, hyperarid Sahara Desert (Green Sahara)[20]. The different timing of changes in temperature[17,18,21] and hydroclimate[21–23] around the AHP allows for disentangling temperature and precipitation effects on $\tau_{soil}$.

Given the absence of proxies for NPP and carbon stock size paleo $\tau_{soil}$ cannot be calculated based on Eq. (1). Instead, we investigate the response of $\tau_{soil}$ to these climatic changes using compound-specific radiocarbon dating (CSRA) of terrigenous biomarkers, i.e. long chain n-alkanoic acids and long chain n-alkanes preserved in marine sediment core GeoB7702-3, which was retrieved in the eastern Mediterranean from the continental margin off the Sinai Peninsula (Fig. 1). Both compounds are constituents of epicuticular leaf waxes and specific biomarkers for higher land plants[24]. In marine sedimentary archives they serve as recorders of terrestrial environmental change[23–25]. At the time of deposition in marine sediments these refractory lipids are commonly pre-aged due to intermediate storage (e.g. in soils) and land-ocean transport[26,27]. The degree of pre-aging (or the age at the time of deposition) is a measure for terrestrial residence times of these compounds which is commonly used to trace changes in terrestrial carbon cycling[16,26,28,29]. Their age at the time of deposition can be determined by radiocarbon dating[26]. However, since soil carbon is a complex mixture of various compounds which all possess different turnover times[30], the ages of leaf-wax lipids only represent a small fraction of the soil organic matter and do not represent $\tau_{soil}$[31]. Ages of leaf-wax lipids generally exceed the calculated mean $\tau_{soil}$ by a multiple[31]. Analyzing the ¹⁴C-ages of n-alkanoic acids in particulate organic matter from a global sample set comprising coastal sediments near river mouths, riverbeds and banks as well as suspension load, ref. 31 identified globally constant offsets between ¹⁴C-ages of n-alkanoic acids and $\tau_{soil}$ (see methods for more details). This allows to calculate catchment-integrating mean $\tau_{soil}$ (in yrs) from the ¹⁴C-ages of n-alkanoic acids in marine sedimentary archives and to monitor changes in the carbon cycle within a river catchment through time.

Here, we deduce past mean $\tau_{soil}$ for the Nile River catchment from the ¹⁴C-ages of leaf-wax biomarkers at the time of deposition at site GeoB7702-3. To calculate the age at the time of deposition of the long chain n-alkanoic acids and long chain n-alkanes we use the "reservoir age offset" notation[32] (given in ¹⁴C years; see methods) between the biomarkers and the atmosphere at the time of deposition (Table 1 and Fig. 2c).

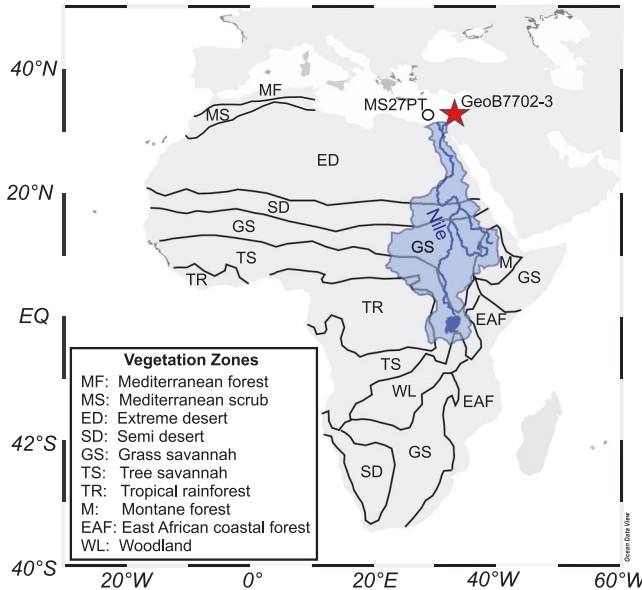

**Fig. 1 | Map of the study area.** African vegetation zones are drawn after ref. 25. The Nile River catchment is marked by the blue shading. The red star indicates the study site GeoB7702-3.

**Table 1 | Reservoir age offsets of leaf-wax lipids, catchment-weighted mean soil carbon turnover times ($\tau_{soil}$), soil mean carbon ages and climate variables for the Nile River watershed over the past 18 kyrs**

| Sample depth [cm] | Deposition age min.–max. [kyrs BP]ᵃ | Deposition age mid-point median [kyrs BP]ᵃ | R n-alkanoic acids [¹⁴C yrs] | R n-alkanes [¹⁴C yrs] | $\tau_{soil}$ [yrs] | Soil mean carbon age [yrs] | δDₚ [‰ VSMOW]ᵇ | T_TEX86 [°C] |
|---|---|---|---|---|---|---|---|---|
| 81.5–84.5 | 1.62–2.29 | 1.93 | 348 ± 240 | 959 ± 146 | 9 ± 6 | 561 ± 392 | −8.8 ± 2.7 | 26.9 ± 0.4 |
| 130–133 | 3.11–3.69 | 3.40 | 733 ± 432 | 1633 ± 167 | 18 ± 11 | 1182 ± 710 | −9.3 ± 2.5 | 26.3 ± 0.6 |
| 198–201 | 5.35–6.01 | 5.70 | 902 ± 331 | 1668 ± 116 | 22 ± 9 | 1455 ± 559 | −8.1 ± 1.5 | 25.1 ± 0.4 |
| 231–234 | 7.24–8.14 | 7.72 | 563 ± 247 | 21 ± 87 | 14 ± 6 | 908 ± 411 | −19.3 ± 5.7 | 26.7 ± 0.7 |
| 251–254 | 9.02–10.11 | 9.66 | 736 ± 196 | 3447 ± 298 | 18 ± 5 | 1187 ± 343 | −27.2 ± 2.1 | 25.3 ± 2.0 |
| 278–281 | 11.05–12.05 | 11.50 | 1631 ± 158 | 3313 ± 178 | 41 ± 6 | 2630 ± 391 | 5.0 ± 2.9 | 19.1 ± 0.7 |
| 297–300 | 12.69–13.73 | 13.21 | 5384 ± 618 | 4334 ± 213 | 134 ± 20 | 8684 ± 1399 | 1.0 ± 2.9 | 20.0 ± 0.7 |
| 359–362 | 16.26–17.07 | 16.67 | 3453 ± 119 | 2415 ± 81 | 86 ± 9 | 5569 ± 657 | 8.3 ± 8.1 | 17.5 ± 1.3 |
| 393–396 | 17.69–18.73 | 18.15 | 8723 ± 212 | 7816 ± 341 | 218 ± 22 | 14069 ± 1625 | 8.3 ± 2.7 | 16.3 ± 0.7 |

Mean $\tau_{soil}$ and soil mean carbon ages are deduced from the reservoir ages offset (R) between n-alkanoic acids and the atmosphere at the time of deposition at site GeoB7702-3 according to ref. 31. R is calculated from compound-specific radiocarbon analysis (CSRA) of the combined n-C₂₆:₀ and n-C₂₈:₀ alkanoic acid homologues (combined measurements and mass-weighted mean, see methods and Supplementary Table 1). As for the n-alkanes R is based on the n-C₂₉, n-C₃₁, n-C₃₃ alkane homologues (combined measurement, see methods). The hydrogen isotopic composition of precipitation (δDₚ) and T_TEX86 are mean values for the range of the deposition age. δDₚ is based on the δD signature of n-alkanoic acids in core GeoB7702-3[23] and given relative to the Vienna Standard Mean Ocean Water (VSMOW). T_TEX86 are sea surface temperature reconstructions at site GeoB7702-3 from ref. 17. The standard deviation (±) is reported along with the results. As for the CSRA results reported as F¹⁴C and Δ¹⁴C the reader is referred to Supplementary Table 1.
ᵃObtained by radiocarbon dating of planktic foraminifera[23].
ᵇCalculated by correcting the δD of the n-C₂₆:₀ and n-C₂₈:₀ alkanoic acids in core GeoB7702-3 for vegetation changes and ice volume[23].

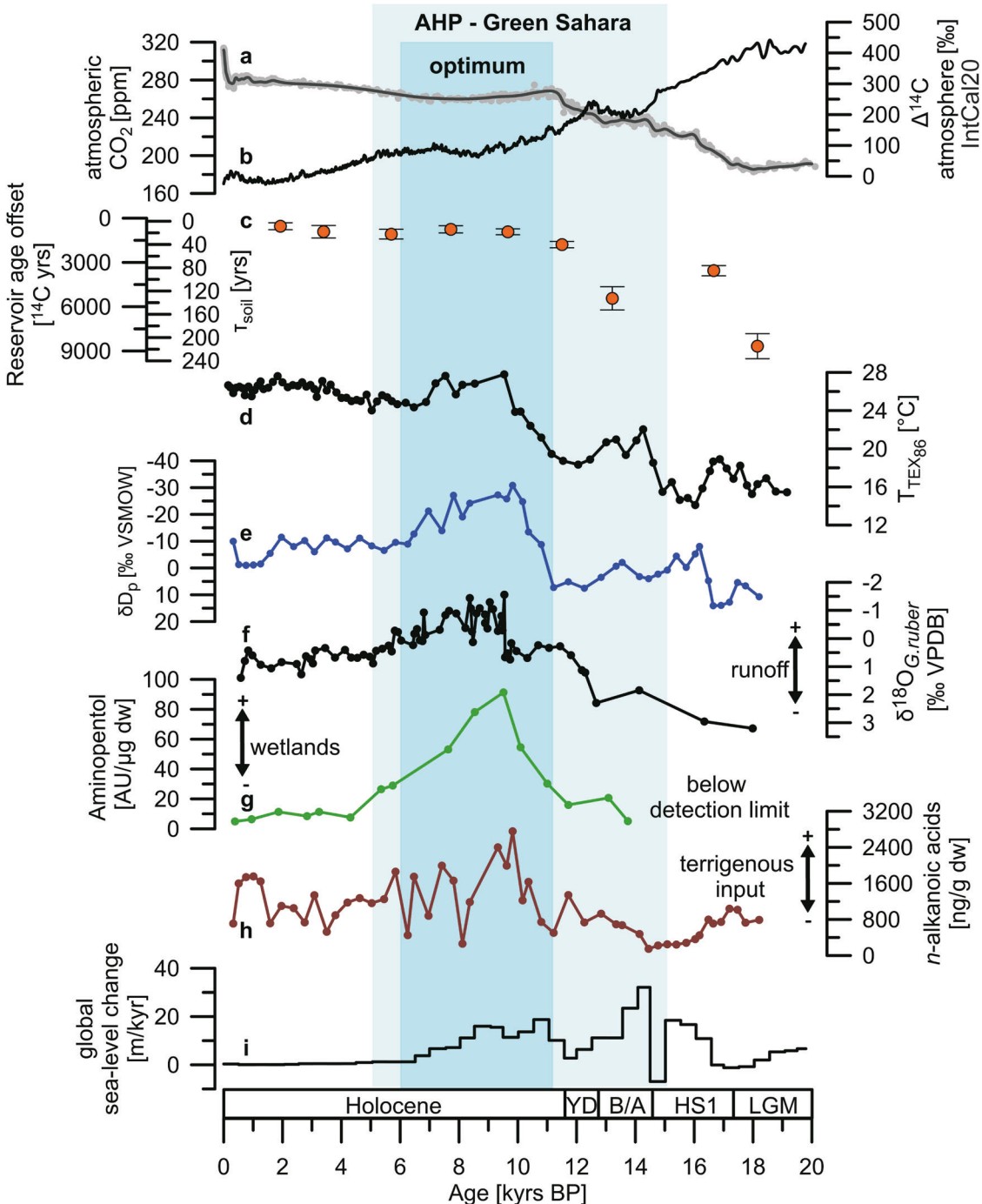

**Fig. 2 | Environmental changes in the Nile-River delta region during the past 18 kyrs. a** Ice-core $CO_2$-contents given as indicator for atmospheric $CO_2$ concentrations (gray dots: data points; black line: spline-smoothed record)[15]. **b** Atmospheric $\Delta^{14}C$ contents according to IntCal20[52]. **c** Reservoir age offsets (R) between the n-alkanoic acids and the atmosphere at the time of deposition at site GeoB7702-3 (this study). $\tau_{soil}$ deduced from R of n-alkanoic acids. Error bars indicate the standard deviation. **d** Sea surface temperature reconstruction for the eastern Mediterranean based on the $TEX_{86}$ proxy from core GeoB7702-3[17]. **e** Hydrogen isotopic composition of precipitation ($\delta D_p$) calculated from the $\delta D$ of n-alkanoic acids from core GeoB7702-3 as proxy for rainfall amount[23]. **f** Oxygen isotopic compositions of the planktic foraminifera species *Globigerinoides ruber* ($\delta^{18}O_{G.ruber}$) in core MS27PT (Fig. 1)

indicating salinity changes in the eastern Mediterranean associated with freshwater runoff from the Nile River[35]. **g** Aminopentol abundances in core GeoB7702-3 used as proxy for the extent of methane-producing wetlands in the catchment (this study). AU: arbitrary units; dw: dry weight of extracted sediment. Additional abundance profiles from the suite of aminobacteriohopanepolyols are given in Supplementary Fig. 1. **h** Concentrations of n-alkanoic acids ($\Sigma n$-$C_{26:0}$, $n$-$C_{28:0}$, $n$-$C_{30:0}$, $n$-$C_{32:0}$) reporting on the land-ocean transport of terrigenous organic matter[23]. **i** Global rate of sea-level change over the last 20 kyrs[33]. The blue bars mark the timing of the African Humid Period (AHP) and Green Sahara and their optimum[19,20]. LGM: Last Glacial Maximum, HS1: Heinrich Stadial 1, B/A: Bølling/Allerød interstadial, YD: Younger Dryas stadial.

## Results and discussion

### Environmental signals in the compound-specific radiocarbon data

The reservoir age offsets of $n$-alkanoic acids and $n$-alkanes in core GeoB7702-3 range between approximately 0 and 8700 [14]C yrs. It is striking that glacial reservoir age offsets (7800–8700 [14]C yrs at 18 kyrs BP) are substantially higher than those during the Holocene (0–3400 [14]C yrs; between ~2–11.5 kyrs BP). This implies a drastic reduction of turnover times of soil carbon during the deglaciation. However, before converting the reservoir age offsets into mean $\tau_{soil}$ three factors that may introduce biases need to be considered.

First, sea level rose by up to 120 m over the deglaciation[33] and coastal erosion during shelf flooding led to the deposition of pre-aged organic matter on continental margins[28,34]. Such processes may mask hinterland signals in the reservoir age offsets of leaf-wax lipids in marine sediments. However, biases from coastal erosion during retrogradation of the Nile Delta are unlikely as the concentration profile of $n$-alkanoic acids in core GeoB7702-3 differs from the global rate of sea-level change[33] (Fig. 2h, i) but resembles the oxygen isotopic composition of planktic foraminifera *Globigerinoides ruber* ($\delta^{18}O_{G.ruber}$) off the Nile River delta, a proxy for freshwater discharge from the Nile River[35] (Fig. 2f). Hence, the export of organic matter was primarily controlled by river runoff[23].

Second, in addition to mineral soils peatlands need to be considered as source of pre-aged organic matter[31]. Anaerobic conditions in wetlands hamper degradation of organic matter leading to its preservation in peat over millennia[36]. During wetland contraction, erosion and fluvial export of this pre-aged material[29] could thus bias the calculations of mean $\tau_{soil}$ of mineral soils[31]. This might be relevant to the Nile River catchment since wetlands occur along the basin today[37]. To constrain wetland dynamics we analyzed a suite of amino-bacteriohopanepolyols (amino-BHPs; Supplementary Fig. 1) which are specific markers for methane oxidizing bacteria in wetlands[38] and thus indicative of the relative extension and contraction of methane producing landcover[29]. Low concentrations of amino-BHPs imply that between 18–11 kyrs BP methane producing permanently flooded wetlands were barely present in the catchment (Fig. 2g and Supplementary Fig. 1) rendering it unlikely that the decrease in the reservoir age offset stems from wetland dynamics. High concentrations of amino-BHPs suggest that wetlands expanded later, i.e. between 11-8 kyrs BP, which probably occurred in response to maximal rainfall and river runoff associated with the AHP-optimum (Fig. 2e, f, g). Contributions of pre-aged organic matter mobilized from wetland contraction at the end of the AHP were probably minor as reservoir age offsets remain constant when amino-BHP concentrations decline in our core (Fig. 2c, g).

Third, river dynamics including morphology and runoff are known controls on the ages of organic matter discharged into the ocean[39,40]. Increased fluvial runoff may strengthen riverbank erosion and export of relatively old material from deeper soil horizons potentially overprinting signals from $\tau_{soil}$[40]. Although the Nile-River runoff increased in response to intensified rainfall during the AHP[22,35] considerable biases from deep-soil erosion are unlikely given the decrease in reservoir age offsets of $n$-alkanoic acids and $n$-alkanes at these times (Table 1, Supplementary Fig. 2). However, intensified Nile River runoff[35] may have increased the transport velocity hampering aging of organic matter during land-ocean transit[39]. This speed-up would have led to smaller ages of plant waxes in core GeoB7702-3 and would be congruent with the observed decrease in our reservoir age offsets. Although signals of the transport efficiency in our data cannot be fully ruled out we consider a predominant control of river dynamics and morphology on ages of discharged organic matter unlikely for the following reasons. River runoff decreased after 7 kyrs BP (Fig. 2f) while the reservoir age offsets of leaf-wax biomarkers remained relatively constant (Table 1; Fig. 2c). The second argument is the similarity between the ages of $n$-alkanoic acids and $n$-alkanes (Table 1 and

Supplementary Fig. 2). As elaborated in ref. 23, $n$-alkanoic acids reflect a local signal from the Nile delta region while the $n$-alkanes provide a catchment-integrating signal[23]. The extensive Nile catchment is characterized by multiple fluvial environments that differ in geomorphology, flow regime and sedimentary processes[41,42]. If such morphologic characteristics exerted substantial control on the ages of organic matter in the fluvial load[39], $n$-alkanoic acids and $n$-alkanes would show different ages and trends which is not the case (Supplementary Fig. 2).

### $\tau_{soil}$ during the past 18 kyrs

Excluding these potential biases, we conclude that reservoir age offsets of the leaf-wax biomarkers in core GeoB7702-3 can be used to calculate mean $\tau_{soil}$ (see methods). For $n$-alkanes the relationship to mean $\tau_{soil}$ is unknown[31] which is why we focus on the $n$-alkanoic acids. Despite the local origin of the $n$-alkanoic acids[23] catchment-wide inferences on changes in $\tau_{soil}$ are justified given the strong similarity with the reservoir age offsets of the $n$-alkanes that provide catchment integrating signals[23] (Supplementary Fig. 2).

During the last 10 kyrs, $\tau_{soil}$ was 9–22 yrs (average 16 yrs) and 218 yrs during the glacial, meaning that $\tau_{soil}$ was reduced by an order of magnitude across the deglaciation (Table 1 and Fig. 2c). $\tau_{soil}$ is regulated by the efflux rates of carbon. Degradation of organic matter via microbial respiration constitutes the majority of the total efflux and contributions of lateral fluxes are minor[13]. As such, the substantial reduction in mean $\tau_{soil}$ attests to a substantial increase in microbial respiration rates over the deglaciation.

It is well constrained that microbial respiration accelerates in response to warming and increased soil moisture[3,7,12]. Both, temperature[17,18,21] and rainfall amount[21–23] increased in the Nile River catchment during the deglaciation (Fig. 2d, e). To investigate the relationship of $\tau_{soil}$ to temperature and rainfall amount we fit the natural logarithm of $\tau_{soil}$ to proxy-based temperature estimates from the eastern Mediterranean[17] and to the hydrogen isotopic composition of paleo precipitation ($\delta D_p$) in the Nile delta[23] (Fig. 3). As mean annual air temperature estimates covering the past 18 kyrs are not available for the Nile River catchment, we use the TEX$_{86}$-based temperature record from GeoB7702-3 interpreted to reflect sea surface temperature (SST) in the eastern Mediterranean[17]. We assume that SST and surface air temperatures in the Nile delta region developed similarly due to heat exchange between the sea surface and the overlying air. As for $\delta D_p$, we use a record based on the $\delta D$ of $n$-alkanoic acids in core GeoB7702-3[23]. $\delta D_p$ is generally controlled by several factors including changes in the moisture source, temperature, evapotranspiration and rainfall amount[43]. In northern Africa and the Mediterranean realm $\delta D_p$ predominantly reflects the amount of rainfall[44].

We find that $\tau_{soil}$ is strongly negatively correlated with temperature ($R^2 = 0.82$; Fig. 3a). A negative correlation of $\tau_{soil}$ with $\delta D_p$ also exists but it is weaker ($R^2 = 0.59$; Fig. 3b). This indicates that temperature was a critical control on microbial respiration rates over the past 18 kyrs (Fig. 3a) while precipitation effects were relatively small. The slope of the correlation in Fig. 3a is a measure for the temperature sensitivity of $\tau_{soil}$ during the past 18 kyrs. The temperature sensitivity of soil respiration and $\tau_{soil}$ is commonly expressed as the $Q_{10}$ value, the factor determining the shift in $\tau_{soil}$ per 10 °C change in temperature[7,45]. $Q_{10}$ is defined as:

$$Q_{10} = e^{10a} \tag{2}$$

where a is the slope of the regression in the temperature-ln($\tau_{soil}$) plot (Fig. 3a). Accordingly, we obtain a $Q_{10}$ of 10.7 (7.0–16.3, 95% confidence interval) for the last 18 kyrs. Note that TEX$_{86}$-based temperatures from core GeoB7702-3 suggest a warming of 10 °C across the deglaciation (Fig. 2d), which is higher than what is typically proposed from other temperature records from the eastern Mediterranean as well as from climate models (3–8°C)[14,17,46]. For a smaller amplitude in deglacial

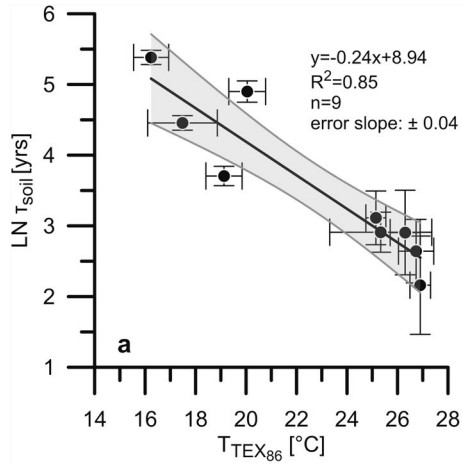

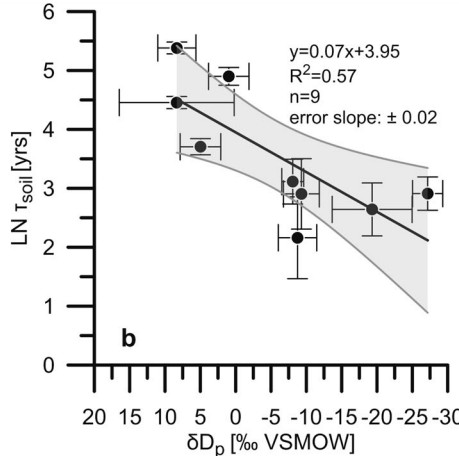

**Fig. 3 | Power–law relationships between $\tau_{soil}$ and temperature and rainfall.**
**a** Correlation with temperature estimates based upon the TEX$_{86}$-proxy (T$_{TEX86}$) from core GeoB7702-3. T$_{TEX86}$ are adopted from ref. [17] and interpreted to reflect sea surface temperature[17]. **b** Correlation with the hydrogen isotopic composition of precipitation ($\delta$D$_p$) which serves as proxy for rainfall amount. $\delta$D$_p$ is calculated from the hydrogen isotopic composition of $n$-alkanoic acids ($n$-C$_{26:0}$ and $n$-C$_{28:0}$ homologues) from core GeoB7702-3[23] and given relative to the Vienna Standard

Mean Ocean Water (VSMOW). In **a** and **b** error bars represent the standard deviation (SD). The gray shadings represent the 95% confidence intervals (CI) and the error of the slope therefore contains 2σ. The p-values for the regressions are <0.05. The temperature sensitivity expressed as the Q$_{10}$-value, i.e. the factor by which $\tau_{soil}$ decreases per 10 °C temperature change[7,45], can be deduced from the slope of the regression line in **a** using Eq. (2) leading to Q$_{10}$ = 10.7 (7.0–16.3, 95% CI).

warming, the slope of the regression line in Fig. 3a would be steeper which would lead to even higher Q$_{10}$ values. Furthermore, our regressions in Fig. 3 are only based on the mean values and uncertainties in y. If with a different regression algorithm also the uncertainties in variables in x direction were considered then the slopes in the regressions would get even steeper.

For modern conditions, Q$_{10}$ values of 1–13 have been reported but mean values commonly are about 2–3 in most biomes[31,47]. Our Q$_{10}$ estimate of 10.7 (7.0–16.3) is at the top of the range substantially exceeding the modern average. Field observations revealed that Q$_{10}$ is spatially and temporally variable and that Q$_{10}$ itself is inversely correlated to temperature[47,48]. That is why ecosystems in colder regions and higher latitudes have relatively high Q$_{10}$ compared to lower latitudes and warm settings[49]. These observations potentially explain why we find rather high Q$_{10}$ for cold glacial and deglacial climates. The dependency of Q$_{10}$ to climate and environmental conditions also indicate that there might not be the rather simple linear relationship between temperature and ln($\tau_{soil}$)[48] which is suggested by the Q$_{10}$ concept, but that the relation between both variables is more complex. If so, our finding of a deglacial (sub-)tropical Q$_{10}$ at the upper end of the observed modern range may also point to a limitation of the Q$_{10}$ concept.

### Implications for the global carbon cycle
The high glacial $\tau_{soil}$ indicate that the carbon exchange between northeastern African soils and the atmosphere was much slower than during the Holocene owing to lower respiration rates during a colder climate. A higher $\tau_{soil}$ agrees with previous estimates of a lower glacial global NPP[50] which is congruent with a lower carbon efflux from soils assuming equilibrium conditions (Eq. (1)). When discussing turnover times of organic carbon in soils and the implications of changes in carbon storage and turnover time for the global carbon cycle one has to acknowledge that soil organic matter is a complex mixture of fast-cycling labile fractions which degrade within years to decades and slow-cycling refractory compounds that decompose on centennial to millennial time scales[30,51]. The assumption that $\tau_{soil}$ determined by the ratio of NPP over carbon stock size (Eq. (1)) is representative of the entire soil carbon pool oversimplifies soil carbon dynamics as the calculation is actually biased towards the fast cycling pool. This

becomes evident when comparing turnover times calculated after Eq. (1) with radiocarbon dates of bulk soil organic matter (the so-called soil mean carbon ages[51]). If soil organic matter was homogenous $\tau_{soil}$ and soil mean carbon ages would match. But in reality $\tau_{soil}$ calculated after Eq. (1) underestimates soil mean carbon ages[51]. This discrepancy is because slow-cycling compounds accumulate in soils owing to their long residence times and dominate the soil organic carbon pool[30,51]. By contrast, $\tau_{soil}$ based on NPP and carbon stock size is biased towards the fast cycling pool as the majority of organic compounds introduced into soils by NPP degrades quickly on years to decades[51]. To investigate the response of soil carbon dynamics to climate change soil mean carbon ages should be considered next to $\tau_{soil}$, in particular because the slow-cycling pool is more vulnerable to climate change than fast cycling compounds[8,31]. It is documented that for a given change in temperature the change in turnover rates is greater for a slow-cycling compounds than for the fast-cycling ones[8]. Given its size, the slow-cycling pool is thus critical for potential positive climate feedbacks from soil carbon dynamics in a warming world[8]. According to refs. [14,31], [14]C-ages of $n$-alkanoic acids off rivers have constant offsets not only with mean $\tau_{soil}$ but also with soil mean carbon ages (integrated over 0–100 cm soil depth; Methods)[31,51]. Calculating soil mean carbon ages from our reservoir age offsets of $n$-alkanoic acids (see methods) reveals that during the last glacial soil organic carbon was up to more than ten thousands of years old (14,000 yrs at 18 kyrs BP; Table 1) which is by an order of magnitude older than during the Holocene (1000 yrs; Table 1). The rejuvenation of soil organic matter accompanying the reduced $\tau_{soil}$ implies a massive mobilization of pre-aged organic carbon from soils during the deglaciation once the climate warmed. Today, respiration constitutes the majority of the total efflux (>90%)[13] and assuming this relation was similar in the past, the decrease in our estimated $\tau_{soil}$ and soil mean carbon ages almost entirely reflects increased efflux of aged CO$_2$ into the atmosphere. Accordingly, the reduction of $\tau_{soil}$ and soil mean carbon ages by an order of magnitude implies an increase in soil-to-atmosphere CO$_2$ flux of a similar size (Eq. 1). This forms a positive feedback to global warming.

During the last deglaciation atmospheric CO$_2$ rose by about 80–90 ppm[15] while the atmospheric radiocarbon content ($\Delta$[14]C) declined concurrently[52] (Fig. 2a, b). To explain these changes oceanic

**Table 2 | $\tau_{soil}$ and soil mean carbon ages for the Ganga–Brahmaputra river catchment during the past 17 kyrs**

| Deposition age [kyrs BP] | n-Alkanoic acid homologues | Mass weighted mean R [$^{14}$C yrs] | $\tau_{soil}$ [yrs] | Soil mean carbon ages [yrs] |
|---|---|---|---|---|
| 0.003 | n-C$_{24:0}$, n-C$_{26:0}$, n-C$_{28:0}$, n-C$_{30:0}$, n-C$_{32:0}$ | 1446 ± 80[a] | 36 ± 4 | 2333 ± 293 |
| 0.004 | n-C$_{24:0}$, n-C$_{26:0}$, n-C$_{28:0}$, n-C$_{30:0}$, n-C$_{32:0}$ | 927 ± 87[a] | 23 ± 3 | 1495 ± 220 |
| 3.54 ± 0.39 | n-C$_{24:0}$, n-C$_{26:0}$, n-C$_{28:0}$, n-C$_{30:0}$, n-C$_{34:0}$ | 7119 ± 1149 | 178 ± 33 | 11483 ± 2261 |
| 6.57 ± 0.42 | n-C$_{24:0}$, n-C$_{28:0}$, n-C$_{30:0}$, n-C$_{32:0}$ | 1489 ± 618 | 37 ± 16 | 2401 ± 1033 |
| 10272 ± 504 | n-C$_{24:0}$, n-C$_{26:0}$, n-C$_{28:0}$, n-C$_{30:0}$, n-C$_{32:0}$ | 3009 ± 749 | 75 ± 20 | 4853 ± 1327 |
| 10.92 ± 0.48 | n-C$_{24:0}$, n-C$_{28:0}$, n-C$_{30:0}$, n-C$_{32:0}$, n-C$_{34:0}$ | 2070 ± 1116 | 52 ± 28 | 3339 ± 1839 |
| 12.74 ± 0.42 | n-C$_{24:0}$, n-C$_{26:0}$, n-C$_{28:0}$, n-C$_{30:0}$, n-C$_{34:0}$ | 3234 ± 1166 | 80 ± 30 | 5216 ± 1971 |
| 13.61 ± 0.23 | n-C$_{24:0}$, n-C$_{26:0}$, n-C$_{28:0}$, n-C$_{30:0}$, n-C$_{32}$, n-C$_{34:0}$ | 1375 ± 830 | 34 ± 21 | 2217 ± 1363 |
| 15.62 ± 0.37 | n-C$_{24:0}$, n-C$_{26:0}$, n-C$_{28:0}$ | 8709 ± 4166 | 217 ± 106 | 14047 ± 6903 |
| 16.77 ± 0.39 | n-C$_{24:0}$, n-C$_{26:0}$, n-C$_{28:0}$, n-C$_{30:0}$, n-C$_{34:0}$ | 6453 ± 2177 | 116 ± 55 | 7506 ± 3612 |
| 16.90 ± 0.10 | n-C$_{24:0}$, n-C$_{26:0}$ n-C$_{28:0}$, n-C$_{30:0}$, n-C$_{32:0}$ | 4004 ± 3507 | 100 ± 88 | 6458 ± 5703 |

$\tau_{soil}$ and soil mean carbon ages (this study) are calculated from compound-specific radiocarbon analysis (CSRA) of n-alkanoic acids in marine sediments from the Bengal Fan (data from ref. 16). CSRA-data of n-alkanoic acids are presented as the reservoir age offset (R) between the n-alkanoic acids and the atmosphere at the time of deposition in the Bengal Fan. R is based upon the mass-weighted results of the listed n-alkanoic acid homologues[16]. The deposition age is adopted from ref. 16. The standard deviations (±) are reported along with the results.
[a]Data of pre-1950 Bengal Fan sediments are adopted from ref. 27.

outgassing of old, $^{14}$C-depleted $CO_2$ (ref. 53) together with contributions from release of aged $CO_2$ from thawing permafrost soils in the Northern Hemisphere have been invoked[28,34,50]. Our findings suggest that, if widespread across the tropics and sub-tropics, the loss of pre-aged carbon from (sub-)tropical soils due to amplified respiration rates may have formed an additional terrestrial source of old $CO_2$ to the atmosphere (Fig. 2a, b) next to the permafrost domain. There is evidence for accelerated soil-carbon turnover in the Ganga-Brahmaputra River catchment as inferred from reservoir age offsets of long-chain n-alkanoic acids from the Bengal Fan[16]. We calculate $\tau_{soil}$ from these data and find that the range of values and the magnitude of deglacial changes ($\tau_{soil}$ falls from ~200 to ~20 yrs; Table 2) are very similar to the results from the Nile River catchment. Thus, given the similarities between datasets from (sub-)tropical river catchments from two continents it is likely that drops in $\tau_{soil}$ by one order of magnitude during Termination I were a common feature across the (sub-)tropics. Interestingly, the radiocarbon data from the Bengal Fan are correlated with rainfall indicating that variability of the Indian summer monsoon played an important role in this positive soil-carbon-climate feedback[16]. However, the results from the Nile River catchment do not confirm the critical involvement of hydroclimate but suggest a direct response of soil respiration rates to warming.

Dynamic global vegetation models (DGVM) allow for investigating the effect of the decreasing $\tau_{soil}$ on the global carbon cycle and atmospheric $CO_2$. We revisit the analysis performed using the Lund Potsdam Jena DGVM (LPJ DGVM)[54] and calculate the differences in $\tau_{soil}$, soil respiration ($R_h$) and soil carbon between the Last Glacial Maximum (LGM; 21 kyrs BP) and pre-industrial conditions (PI; 1 kyr BP). The results are shown in Fig. 4a–e. Details of the simulation are described in the methods and ref. 54.

As described in ref. 54, the model simulates a total increase in the global terrestrial carbon pools of 820 PgC between the LGM and PI[54]. This agrees well with the median of 850 PgC estimated by a recent multi-proxy approach[55] showing that the simulated global patterns agree with other studies. When subtracting the effect of $CO_2$ fertilization, the model suggests a reduction of the global land carbon stock by 200–250 PgC for areas unaffected by rising sea level or ice retreat for PI relative to the LGM[54]. This represents the summed-up change in vegetation and soil carbon caused by temperature and precipitation variability and is attributed to higher global turnover rates at PI[54]. However, we find pronounced discrepancies between our data-based reconstruction of the change in $\tau_{soil}$ (decrease by 200 yrs, Table 1) and the simulated values for the wider (sub-)tropics (Fig. 4a, b). The model indicates marginal change in $\tau_{soil}$ of less than 50 yrs. Substantial

changes of similar magnitude as in our reconstruction are simulated only in the northern high latitudes (Fig. 4a, b). Considering the relationships in Eq. (1), the underestimation of changes in (sub-) tropical $\tau_{soil}$ translates into underestimated, simulated changes in microbial respiration rates, respectively $CO_2$ efflux. The discrepancies between our data-based estimates of $\tau_{soil}$ and the LPJ DGVM simulations suggest that the climate feedback from amplified (sub-)tropical soil respiration due to the deglacial warming is underestimated in models.

The temperature sensitivity of $\tau_{soil}$ is the key parameter for estimating changes in the soil carbon content in response to warming. Some models operate with constant $Q_{10}$ values, typically 2 (refs. 5,56), but in DGVMs the relationship between temperature and soil respiration is typically described with a rather complex equation. For example, the dependency embedded in the LPJ DGVM[57] is, when plotted as relative loss of soil carbon content versus temperature change, dependent on the baseline temperature $T_0$, from which the anomalies are calculated. Results are for a $T_0$ of 10, 20, or 30 °C similar to a $Q_{10}$ of 3, 2.3, or smaller than 2.0, respectively (Fig. 5). This pronounced difference to our data-based estimate of $Q_{10} = 10.7$ (7.0–16.3) probably explains at least in parts why the simulated changes in $\tau_{soil}$ between LGM and PI are substantially smaller than in our data-based reconstructions. However, since the data from the Indian subcontinent point to a stronger influence of precipitation on $\tau_{soil}$ there[16], but the simulated $\tau_{soil}$ in the LPJ DGVM results are not different between Africa and India (Fig. 4a, b) some other substantial shortcomings possibly exist in the model, which we cannot identify here. Discrepancies between simulations and data-based estimates of modern $\tau_{soil}$, respectively terrestrial ecosystem respiration have also been documented previously[4,58] and have been attributed to inaccurate parameterizations of $Q_{10}$ (ref. 58).

Our study provides data-based evidence for a reduction in mean soil carbon turnover time and soil mean carbon ages by an order of magnitude in (sub-)tropical Africa during the last deglaciation. These results suggest that carbon sequestration via vegetation and soils was slower but the efficiency of the soils to remove carbon from the atmosphere and to protect it from biogeochemical cycling was higher. We conclude that microbial respiration rates amplified in direct response to rising temperature and that the release of pre-aged $CO_2$ from (sub-)tropical soils into the atmosphere may have contributed to rising atmospheric $CO_2$ and declining atmospheric $\Delta^{14}$C, a mechanism that has not received much of attention so far. However, for a thorough assessment of the impacts on the global carbon cycle more data-based reconstructions across the (sub-)tropics are needed to obtain a comprehensive view on the timing and

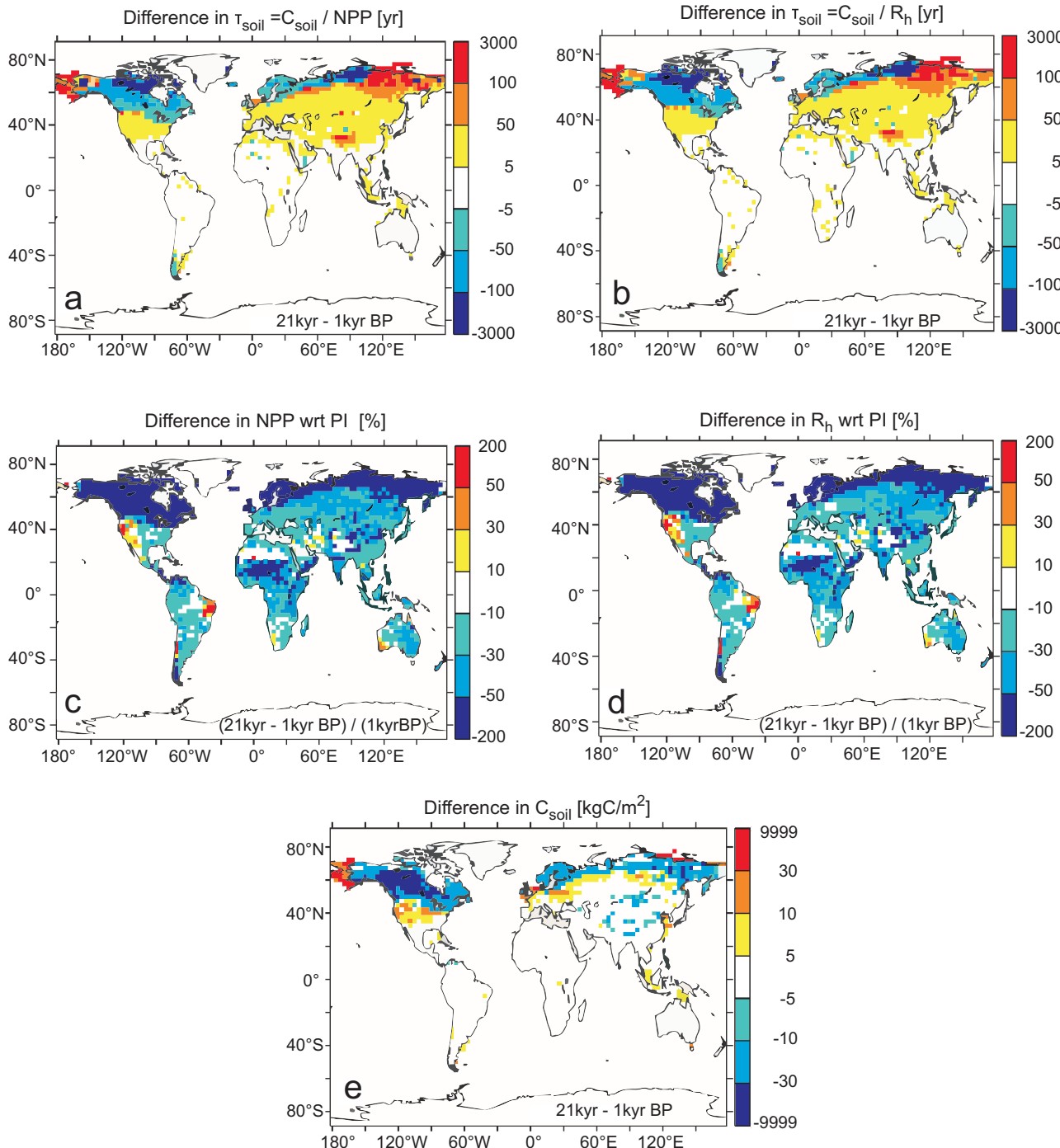

**Fig. 4 | Recalculation of results from the Lund Potsdam Jena Dynamic Global Vegetation Model (LPJ DGVM) over the last 21 kyrs.** These LPJ results are from simulations identical to those that have been forced by the Hadley center climate model as discussed in ref. 54. Relative changes between the LGM and pre-industrial conditions (PI, here: 1 kyr BP) are shown. **a** $\tau_{soil}$ calculated based on the carbon influx (net primary production (NPP)). **b** $\tau_{soil}$ based on the carbon efflux ($R_h$), where $R_h$ is the heterotrophic respiration. Large positive anomalies (red) occur on shelf areas inundated during deglacial sea-level rise, while the areas with large negative anomalies (blue) were covered by large continental ice sheets during the LGM. Calculating $\tau_{soil}$ from net primary production (NPP) reveals similar results as the calculation from respiration fluxes ($R_h$) indicating that NPP and $R_h$ are in equilibrium. **c** Relative changes in NPP. **d** Relative changes in $R_h$. **e** Absolute changes in soil carbon ($C_{soil}$).

magnitude of changes in $\tau_{soil}$ and to evaluate the role of soil-carbon feedbacks outside the permafrost domain during the deglaciation. Moreover, the disagreement between our data and the LPJ DGVM simulations stresses that more research on temperature sensitivity of soil carbon turnover under different settings and different changing climatic boundary conditions is necessary to bring reconstructions and models in closer agreement.

## Methods

### Core material and chronology

Gravity core GeoB7702-3 was retrieved onboard RV Meteor at the continental slope off the Sinai Peninsula during cruise M52/2 in 2002[59]. Due to the anticlockwise surface circulation in the eastern Mediterranean the fluvial load of the Nile River is transported eastward along the coast so that terrigenous biomarkers in core GeoB7702-3 serve as

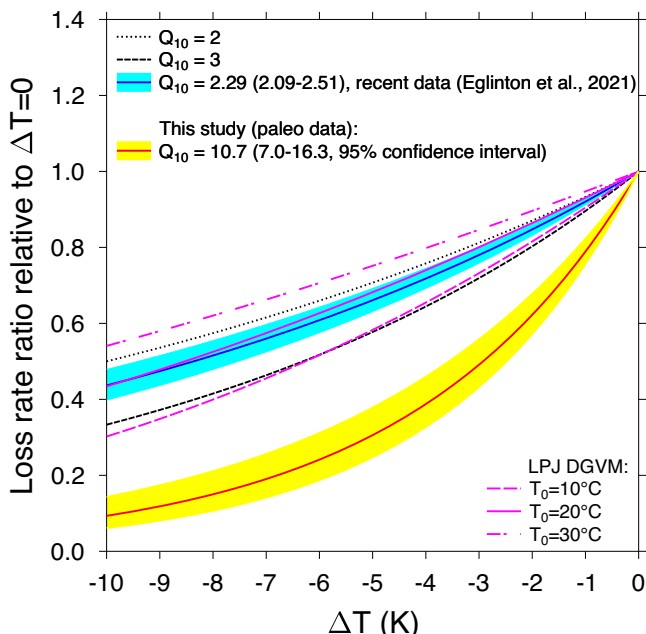

**Fig. 5 | Transferring our temperature-dependent soil carbon turnover time into the Q₁₀ concept.** The relative carbon loss ratio ($f/f_0$, where $f_0$ is the efflux at $\Delta T = 0$) as function of temperature anomaly is plotted for different $Q_{10}$, including results based on recent data by Eglinton et al.[31]. In addition the output of this soil carbon loss rate for the equation used in the LPJ DGVM is plotted for anomalies for three different temperature baselines (Eq. (23) in ref. 57).

recorders of environmental change in the Nile River watershed[17,22,23]. Prior to sample preparation, the core was stored at 4 °C. The sample set for bacteriohopanepolyol (BHP) quantification comprised 21 samples. Samples for compound-specific radiocarbon analysis (CSRA) were taken from 9 selected horizons (~2 cm thickness). Age depth modeling is based upon 24 radiocarbon dates of planktic foraminifera and was previously published in ref. 17 and updated by ref. 23.

**Lipid extraction**
Samples were freeze-dried and homogenized with a mortar. Samples for CSRA (ca. 100–120 g) were extracted with Dichloromethane (DCM):Methanol (MeOH) 9:1 (v/v) using a Soxhlet-apparatus (60 °C, 48 h) and were processed without internal standards. The samples were hydrolyzed with 0.1 N potassium hydroxide (KOH) in MeOH:H₂O 9:1 (v/v) at 80 °C for two hours. Neutral compounds were extracted with *n*-hexane, acids with DCM after acidifying the saponified solution with hydrochloric acid (HCl). Hydrocarbons were separated from polar compounds by column-chromatography using deactivated SiO₂. The hydrocarbons were eluted with *n*-hexane, polar compounds with DCM:MeOH 1:1 (v/v). The fatty acids were derivatized to fatty acid methyl esters (FAME). The methylation was performed with MeOH of known Δ¹⁴C, together with HCl at 50 °C. Air in the headspace of the sample-tube was replaced by nitrogen gas (N₂). FAMEs were recovered with *n*-hexane and were subsequently cleaned-up with column chromatography using deactivated SiO₂ and NaSO₄. FAMEs were eluted with DCM:Hexane 2:1 (v/v).

Freeze-dried sediment samples dedicated for BHP analysis (ca. 3–6 g) were extracted using a modified Bligh and Dyer extraction[60]. The sediment samples were ultrasonically extracted (10 min) with a solvent mixture containing MeOH, DCM and phosphate buffer (2:1:0.8, v:v:v). After centrifugation, the solvent was collected, combined and the residues re-extracted twice. The combined solvent layers were added to separatory funnels and separated from the aqueous layer by the addition of DCM and Milli-Q water. After the layers separated, the

bottom layer (DCM) was drawn off and collected, while the remaining aqueous layer was washed twice with DCM. The combined DCM layers were dried under a continuous flow of N₂. Aliquots of the total lipid extracts (TLEs) were obtained and DGTS (1,2-dipalmitoyl-sn-glycero-3-O-4′-(N,N,N-trimethyl)-homoserine, Avanti Polar Lipids) was added as an internal standard before ultra-high performance liquid chromatography – ultra high resolution mass spectrometry (UHPLC-HRMS) analysis.

**UHPLC-HRMS analysis of non-derivatized BHPs**
Non-derivatized BHPs were quantified by injecting 1% of the TLE with 2 ng internal standard (DGTS) dissolved in MeOH:DCM (9:1, v:v) on a Dionex Ultimate 3000RS ultra-high performance liquid chromatography (UHPLC) system connected to a Bruker maXis Plus Ultra-High Resolution quadrupole time-of-flight tandem mass spectrometer (UHR-qTOF-MS) equipped with an ESI ion source operating in positive mode (Bruker Daltonik, Bremen, Germany). The non-derivatized BHP analysis was performed according to ref. 61 with a column temperature of 30 °C and a modified separation method. Briefly, separation was achieved on an Acquity BEH C18 column (2.1 × 150 mm, 1.7 μm particle size, Waters, Eschborn, Germany) and a solvent system consisting of eluent A of MeOH:H₂O (85:15) and eluent B MeOH:isopropanol (1:1) with both containing 0.12 % (v/v) formic acid and 0.04 % (v/v) aqueous ammonia. Compounds were eluted with 5% B for 3 min, followed by a linear gradient to 60% B at 12 min and then to 100% B at 50 min and holding at 100% B until 80 min. The column was then equilibrated for 20 min leading to a total run time of 100 min. The flow rate was held constant at 0.2 ml min⁻¹. Mass spectra were acquired in positive ion monitoring of *m/z* 50 to 2000 and data-dependent fragmentation of the most abundant ions (dynamically selected, typically 3–8) for a total cycle time of 2 s and dynamic exclusion (activation after 5 spectra, release after 15 s). Ion source settings and parameters for detection and fragmentation of BHPs were optimized while infusing extracts. Every analytical run was mass-calibrated by loop-injection of Agilent ESI-L tune mix and lock mass calibration (*m/z* 922.0098, added in ESI source) of each mass spectrum, leading to typical mass deviations of <1–3 ppm.

BHPs were identified based on the exact mass of the protonated or ammoniated molecular ion, relative retention time and MS² fragmentation similar to ref. 61. Extracted ion chromatograms (EIC) of the most abundant molecular ion (10 mDa mass accuracy window) were used to (semi-)quantify individual BHPs by peak integration. MS variability and ion suppression was controlled by the peak area of the DGTS internal standard. As no authentic standards were available for BHP quantification, abundances are reported based on peak areas of the individual BHPs normalized to the dry weight of the extracted sediments (i.e., in arbitrary units (AU)/μg dw).

**Purification of leaf-wax lipids**
For CSRA the target FAMEs and *n*-alkanes were purified using preparative capillary gas chromatography[62]. The purification was performed on an Agilent 7890B gas chromatograph (GC), equipped with a temperature programmable cooled injection-system (CIS, Gerstel) and connected to a preparative fraction collector (PFC, Gerstel). Separation was performed on a Restek Rxi-1ms fused silica capillary column (30 m, 0.53 mm i.d., 1.5 μm film thickness). All samples were injected repeatedly with 5 μL per injection from a concentration of 1 μg/μl (FAMEs) and 500 μg/μl (*n*-alkanes) using *n*-hexane. The injector was operated in solvent vent mode (vent: 100 ml/min, 0 psi until 0.12 min). The CIS temperature program was: 60 °C (0.05 min), 12 °C/s to 320 °C (5 min), 12 °C/s to 340 °C (5 min). The GC temperature program was set: 60 °C (2 min), 20 °C/min to 150 °C, 8 °C/min to 320 °C (40 min). Helium was used as carrier gas (4.0 ml/min). The transfer line and PFC were heated at 320 °C while the traps for collection were maintained at room temperature. The backflush system of the PFC was constantly switched off. The traps were rinsed with *n*-hexane to recover the

purified compounds. Splits (0.1%) were analyzed by GC-FID to check for potential contaminants and to quantify the purified target compounds for CSRA.

## CSRA

The isotopic ratio ($^{14}$C/$^{12}$C) of the FAMEs and $n$-alkanes was determined by Accelerator Mass Spectrometry (AMS). The measurements were carried out on the Ionplus MICADAS-system equipped with a gas-ion source[63–65] at the Alfred Wegener Institute Helmholtz Centre for Polar and Marine Research, Bremerhaven. CSRA was performed according to the protocols described in ref. [66]. In short, the purified individual target compounds were transferred into tin capsules and packed. As for FAMEs, the $n$-C$_{26:0}$ and $n$-C$_{28:0}$ homologues were prepared individually except for two samples for which the homologues had to be combined in order to achieve adequate sample size (Supplementary Table 1). For $n$-alkanes we combined the $n$-C$_{29}$, $n$-C$_{31}$ and $n$-C$_{33}$ homologues to obtain enough material for dating. Samples were combusted via the Elementar vario ISOTOPE EA (Elemental Analyzer) and the produced $CO_2$ was directly transferred into the coupled MICADAS. Radiocarbon contents of the samples were analyzed along with reference standards (oxalic acid II; NIST 4990c) and blanks (phthalic anhydride; Sigma-Aldrich 320064) and in-house reference sediments. In order to account for $^{13}$C isotopic fractionation, the $^{14}$C/$^{12}$C by convention is normalized to a δ$^{13}$C value of −25‰ PDB, the postulated mean value of terrestrial wood[67]. Blank correction and standard normalization were performed via the BATS software[68]. The AMS results are reported as "fraction modern carbon" (F$^{14}$C) and Δ$^{14}$C as defined in ref. [67].

## Assessment of procedure blanks and correction

To correct for carbon introduced during sample processing, procedure blanks were assessed by isolating $n$-alkanoic acids from a modern and a fossil standard material according to the methods described above. Leaves of a corn plant, collected in 2019, were used as modern standard (F$^{14}$C: 1.0096 ± 0.0024) while "Rekord" coal-briquette (lignite from Lusatia, Eastern Germany) served as fossil standard (F$^{14}$C: 0.0019 ± 0.0002). For the coal, asphaltene precipitation was performed additionally using DCM:MeOH 97:3 (v/v) and pentane. The mass and the F$^{14}$C of the procedure blank were assessed using a Bayesian approach according to ref. [69]. The blank had a mass of 3.079 ± 0.433 μgC with an F$^{14}$C of 0.529 ± 0.072. Blank-correction of the samples and error propagation was performed using mass balance. The blank corrected F$^{14}$C-values of FAMEs were further corrected for the methyl-group, which had been added during the derivatization process, using isotopic mass balance.

## $^{14}$C-ages of the lipids at the time of deposition

The age of the compounds at the time of deposition can be calculated using the "reservoir age offset" ($R$)[32] which describes the age offset (in $^{14}$C years) between two carbon reservoirs at a given time[32]. In our case it was calculated from the ratio of the radiocarbon contents of the sample and the atmosphere at the time of deposition in marine sediments (Eq. (3)).

$$R = 8033 \times \ln\left(\frac{F^{14}C_{initial}}{F^{14}C_{atm}}\right) \qquad (3)$$

where F$^{14}$C$_{initial}$ is the F$^{14}$C-value the sample had at the time of deposition at site GeoB7702-3 and F$^{14}$C$_{atm}$ is the radiocarbon content of the atmosphere. F$^{14}$C$_{initial}$ can be calculated by correcting the measured F$^{14}$C-value of the sample (F$^{14}$C$_{sample}$) for the decay that has taken place since the deposition (Eq. (4)).

$$F^{14}C_{initial} = F^{14}C_{sample} \times e^{\lambda t} \qquad (4)$$

where t is the time of deposition and λ the decay constant of radiocarbon[67]. The time of deposition was inferred from radiocarbon dates of planktic foraminifera (core chronology)[23]. F$^{14}$C$_{atm}$ values were adopted from IntCal20[52]. In case of samples for which the F$^{14}$C values of the $n$-C$_{26:0}$ and $n$-C$_{28:0}$ homologues had been measured separately, we calculated $R$ from the abundance-weighted mean of the F$^{14}$C values in order to keep comparability with samples for which the two homologues had been combined prior to AMS measurement (Supplementary Table 1).

## Calculation of $\tau_{soil}$ and soil mean carbon ages

The authors of ref. [31] discovered a linear relationship between the $^{14}$C-ages of long-chain $n$-alkanoic acids and catchment-weighted mean $\tau_{soil}$ in a global dataset of near-coastal sediments, suspended coastal sediments near river mouths, riverbeds and banks as well as suspension load (Eq. (5)).

$$Age_{n-alkanoic\,acid} = 40.1 \times \tau_{soil} \qquad (5)$$

where the age$_{n\text{-alkanoic acid}}$ is given in $^{14}$C years[31]. Under the premise that this relationship has remained constant since the last glacial, we calculated $\tau_{soil}$ from Eq. (5) using the reservoir age offsets at the respective time of deposition at site GeoB7702-3.

In ref. [31], constant offsets between $n$-alkanoic acids and soil mean carbon ages have been reported (Eq. (6)).

$$Age_{n-alkanoic\,acid} = 0.62 \times soil\,age \qquad (6)$$

The soil mean carbon age here is defined as the age integrated over the top 100-cm depth[31,51]. Age$_{n\text{-alkanoic acid}}$ is the $^{14}$C-age[31].

Using Eqs. (5), and (6), we calculated paleo-$\tau_{soil}$ and soil mean carbon ages from the reservoir ages offsets ($R$; Eq. (3)) of $n$-alkanoic acids in core GeoB77023.

The sample set of ref. [31] covers a broad range of latitude (73 °N to 38 °S) and consequently represents different biomes and climate zones from tropical rainforest to arctic tundra. It reflects broad ranges of annual air temperature (−16 to 27 °C) and mean annual precipitation (amount 230–2200 mm/yr)[31]. The range of $^{14}$C-ages from $n$-alkanoic acids covered by the dataset is recent to >10,000 yrs[31]. The ages at the time of deposition calculated for the $n$-alkanoic acids in core GeoB7702-3 are within that range (348 ± 240 to 8723 ± 212 yrs; Table 1 and Supplementary Table 1). Thus, our inferred $\tau_{soil}$ are within the calibrated range. Since the relationship between $\tau_{soil}$ and the ages of $n$-alkanes at the time of deposition is unknown, we cannot convert our $n$-alkane age into $\tau_{soil}$.

## Dynamic global vegetation model simulation

Temperature and soil moisture effects have been implemented in dynamical global vegetation models for decades[45,57]. For this study, we revisited the analysis performed by refs. [54,70] using the LPJ DGVM and investigate changes in $\tau_{soil}$, net primary production (NPP), soil respiration ($R_h$) and soil carbon between the Last Glacial Maximum (LGM; 21 kyrs BP) and pre-industrial (PI, 1 kyrs BP; Fig. 4). The global land carbon cycle was transiently simulated across Termination I subtracting the effect of $CO_2$ fertilization and restricting the analysis to areas unaffected by rising sea level or continental ice retreat[54]. For this study, $\tau_{soil}$ is calculated according to Eq. (1) using the simulated soil–carbon stock and the simulated NPP and $R_h$, respectively.

## Data availability

The biomarker and radiocarbon data generated in the study have been deposited in the PANGAEA database under the following: https://doi.org/10.1594/PANGAEA.973255; https://doi.org/10.1594/PANGAEA.973253; https://doi.org/10.1594/PANGAEA.973254.

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

## Acknowledgements
The study was funded by the DFG Cluster of Excellence: "The Ocean Floor—Earth's Uncharted Interface" (EXC-2077, Project 390741603) which was granted to MARUM/University of Bremen. We are grateful to Jürgen Pätzold for providing sample material of core GeoB7702-3. We thank Ralph Kreutz for support during sample processing for CSRA and GC-maintenance. Julia Cordes is acknowledged for technical support during BHP analysis. Pushpak Nadar is thanked for assistance during sample processing for BHP analysis and core sampling for CSRA. Hendrik Grotheer is thanked for support during the AMS measurements at AWI.

## Author contributions
V.D.M. and E.S. developed the concept of the study supported by B.W., G.M. and P.K. V.D.M. carried out the sample preparation and data analysis in the laboratories and performed data processing. N.T.S. and J.L. conducted the analysis of Bacteriohopanepolyols. P.K. performed the simulations with the LPJ DGVM. All authors were involved in the interpretation and discussion of the results. V.D.M. drafted the manuscript with contributions from all co-authors.

## Funding

## Competing interests
The authors declare no competing interests.
