## [Peer Review file · Nature Communications]

Dominant control of temperature on (sub-)tropical soil carbon turnover

Corresponding Author: Dr Vera Dorothee Meyer

Version 0:

Reviewer comments:

Reviewer #1

(Remarks to the Author)

The paper presents analysis of radiocarbon dating of carbon compounds in marine sediment cores to infer the average residence time and age of soil carbon in the supplying catchment. The data is retrieved from the Nile River delta and spans the last deglaciation and Holocene. The analysis of the authors suggests drastic changes in residence time/turnover over this time frame, exceeding those of current land surface models. This in turn is relevant as soil carbon turnover is a major determinant of the trajectory of soil carbon under global change, possibly creating large positive feedbacks in the climate system under anthropogenic warming.

I believe the method and the analysis is intriguing. I admit to understanding little of technical part of the sediment analyses, the lipid compounds, and their relationship with the bulk soil carbon and its soil turnover, and hope that another reviewer can provide an assessment. My expertise lies in the biogeochemistry of the carbon cycle, and their representation in land surface models.

I have a few comments that I think need to be addressed or at least would help me to gain some clarity.

1) My confusion about age: The author refer to age in different way. At times it is the time until present and sometimes it is the pre-depositional age. I hope the authors find a way to clarify this. For example use 'pre-depositional age' (or a similar expression) consequently. Similarly, could you explain the difference between turnover and age for the benefit of the reader (and me 😊). The formula for turnover is the mean residence time and therefore represents also the mean age of the soil, but clearly here and in ref 30 in the paper they are different.

2) Presentation of turnover vs. temperature: Many models use a Q10 formulation, an exponential function. Hence the plot of $\ln(\tau)$ vs. temperature would reveal a Q10 (the

3) slope of such a plot is $\ln(Q10)/10$. Hence the Q10 could be inferred (about 11 here, if I am correct) which is way higher than in models (about 2-3), including LPJ.

4) Offset and c3/c4 vegetation transition. Glacial/interglacial change with distinct temperature, moisture and CO₂ changes creates shifts in vegetation types, and particularly in in c3 vs. c4 photosynthetic pathway, which is especially pronounced in warm areas with seasonal precipitation (i.e. subtropical). With fractionation against 14C being very different between C4 and C3, this could considerably affect offset? Perhaps this was considered in this study, but I don't see it discussed here. If omitted, a 20 permil fractionation difference would (according to my -perhaps flawed- calculation) translate into a an about 160 years offset, which would be in the vicinity of the differences in turnover times reported here. If 13C was measured along 14C (maybe it was measured but not shown?), maybe this could help disentangle this effect?

5) Error propagation: Does the calculation of tau include potential errors for equation 3 and 4? The relationship between lipid age and tau has some error to, and so it may be propagated in this work.

Reviewer #2

(Remarks to the Author)

Review and revise the manuscript

Review conclusion: Acceptance after major revision

The article uses the radiocarbon dating method derived from plant derived lipids, combined with temperature and rainfall reconstructed from sediment cores of the eastern Mediterranean that received terrestrial materials from the Nile River Basin, to investigate the dominant control of temperature on carbon turnover in subtropical and tropical soils over the past 18000

years. The research work is solid, the content is rich, the methods are advanced, and it has good innovation. I am very interested in this part of the content, but it is not easy to read the article. The most important issue is that the main research question and conclusion are not very clear, and the writing logic may need to be reorganized for readers to better understand the content. The second question is whether the definition of carbon in the article is reasonable and accurate, which is very important. At least in my opinion, I don't understand what redefining carbon means? And how to determine, does it refer to the rate or quantity of carbon outflow and inflow, how to calculate it is not clearly stated in the article? Then I still don't understand why the reservoir age shift of leaf wax biomarkers can be used to calculate average soil carbon. This part may require more explanation and clarification. The research is very interesting and meaningful, but there are still some issues that need to be modified and resolved. The specific opinions are as follows:

1. What is the meaning of redefining carbon in Line 30? Can't we directly use organic carbon content? How to determine f, does it refer to the rate or quantity of carbon outflow and inflow, how to calculate it, and what is the unit?
2. Why Lines 33-35 said that the impact of water climate on soil carbon turnover time plays a strong controlling role in low latitude climate, more important than temperature? I don't understand. Is there any theoretical support?
3. Lines 49-55: The two biomarkers (e.g., long-chain n-alkanes and n-alkanoic acids) were a bit suddenly proposed here, I suggest to add some descriptions for explaining why you choose these two biomarkers? As I am studying modern processes and mechanisms of these two biomarkers for more than ten years, I totally support that you use these two biomarkers to demonstrate the responses of soil carbon cycle to climate changes, but more explanations are needed here. Please refer some references (e.g., n-alkane for Liu and An, 2020 and Liu et al., 2022; ESR; n-alkanoic acids for Liu et al., 2024; SCES).
4. Line 107-108: Why n-alkanoic acids reflect a local signals while n-alkane provide a more catchment-integrating signal? We know these two biomarkers originate the same precursor (i.e., acyl-ACP).
5. Does the topic consider adding marine sediments? The research is not focused on traditional soils.
6. At the beginning, the research site was in the Nile Delta region, and later it moved to tropical and subtropical regions as well as global analysis. I don't know how this changed in between. Can the Nile Delta region represent subtropical regions?
7. I didn't understand the part about the time in the article. It started in the environmental changes in the Nile Delta region in the previous 18 years, and then it was before glaciers until the New Century. How did it change during this period? What did the environmental changes in the Nile Delta region in the previous 18 years mean? What is the connection to the other content? I don't quite understand?
8. In my opinion, Lines 124-129 has no theoretical support.
9. Lines 135-136: the hydrogen isotopic composition of paleo precipitation (δD_p), serve as a common proxy for the amount of rainfall? This needs a detailed explanation, because our modern investigation showed that δD_p is mainly controlled by temperature at the global scale, instead of rainfall amount, only in some intensive monsoon regions, the δD_p is dominately controlled by rainfall amount, so please supplement some modern data for supporting this inference.
10. Why do you say that in Lines 162-163? Is there any data support for non-tropical and subtropical areas? Doesn't this change exist in non-tropical and non-subtropical regions?
11. Should different vegetation zones in Figure 1 be marked with different colors.
12. It would be better to mark the year on the horizontal axis in Figure 2.
13. It seems unreasonable to use surface temperature instead of sea surface temperature in the eastern Mediterranean in Figure 3.

REVIEWER COMMENTS

Reviewer #1 (Remarks to the Author):

The paper presents analysis of radiocarbon dating of carbon compounds in marine sediment cores to infer the average residence time and age of soil carbon in the supplying catchment. The data is retrieved from the Nile River delta and spans the last deglaciation and Holocene. The analysis of the authors suggests drastic changes in residence time/turnover over this time frame, exceeding those of current land surface models. This in turn is relevant as soil carbon turnover is a major determinant of the trajectory of soil carbon under global change, possibly creating large positive feedbacks in the climate system under anthropogenic warming.

I believe the method and the analysis is intriguing. I admit to understanding little of technical part of the sediment analyses, the lipid compounds, and their relationship with the bulk soil carbon and its soil turnover, and hope that another reviewer can provide an assessment. My expertise lies in the biogeochemistry of the carbon cycle, and their representation in land surface models.

I have a few comments that I think need to be addressed or at least would help me to gain some clarity.

Dear reviewer,

Thank you very much for your positive feedback on our manuscript and for considering it a relevant piece of work. We feel that your comments are very constructive and help to improve our work.

Below, we specify how we address your points in the revised version of our manuscript and we hope that clarity improved by the changes made. The lines numbers given refer to the track-changes version of the manuscript.

1) My confusion about age: The author refer to age in different way. At times it is the time until present and sometimes it is the pre-depositional age. I hope the authors find a way to clarify this. For example use 'pre-depositional age' (or a similar expression) consequently. Similarly, could you explain the difference between turnover and age for the benefit of the reader (and me 😊). The formula for turnover is the mean residence time and therefore represents also the mean age of the soil, but clearly here and in ref 31 in the paper they are different.

Reply: You are right saying that turnover time means residence time of organic matter in soils and theoretically should equal the mean age of the soil. However, turnover time assumes that all carbon is cycling homogenously, which oversimplifies soil carbon dynamics and the complexity of the soil organic carbon. It has been shown that the estimate of residence time based on radiocarbon dating of bulk soil organic carbon is higher than calculations based on the steady-state assumption and using the ratio of carbon stock size over NPP (τ_{soil} ; eq.1 in our manuscript) as documented e.g. in Shi et al. (2020)(ref. 51). The reason is as follows: Organic matter is a complex mixture of fast cycling, labile fractions and slow-cycling refractory fractions. That means some decompose very quickly within years or decades while other refractory components survive over millennia. Therefore, slow-cycling components accumulate in soils dominating the soil organic carbon pool. Accordingly, the age of soil organic carbon determined by radiocarbon dating represents the older, slow cycling fractions. By contrast, the calculation of τ_{soil} using NPP and the carbon stock size is biased towards fast cycling components as "most net primary production cycles through relatively small soil carbon pools on timescales of years to decades" as note by Shi et al. (2021).

We added a short paragraph to lines 238-263 to clarify this.

2) Presentation of turnover vs. temperature: Many models use a Q10 formulation, an exponential function. Hence the plot of $\ln(\tau)$ vs. temperature would reveal a Q10 (the slope of such a plot is $\ln(Q_{10})/10$). Hence the Q10 could be inferred (about 11 here, if I am correct) which is way higher than in models (about 2-3), including LPJ.

Reply: That is a good point. We are aware of the Q_{10} formulation commonly used to express temperature sensitivity. You are correct that Q_{10} inferred from our data is 11 and is higher than the values most models operate with. We decided that this is a relevant and intriguing observation and the revised version contains a short discussion of Q_{10} (lines 200-228 and 312-328). An additional figure is also provided to demonstrate the effect of different Q_{10} on the carbon loss rate from soils (Figure 5).

4) Offset and c3/c4 vegetation transition. Glacial/interglacial change with distinct temperature, moisture and CO_2 changes creates shifts in vegetation types, and particularly in in c3 vs. c4 photosynthetic pathway, which is especially pronounced in warm areas with seasonal precipitation (i.e. subtropical). With fractionation against ^{14}C being very different between C4 and C3, this could considerably affect offset? Perhaps this was considered in this study, but I don't see it discussed here. If omitted, a 20 permil fractionation difference would (according to my -perhaps flawed- calculation) translate into a an about 160 years offset, which would be in the vicinity of the differences in turnover times reported here. If ^{13}C was measured along ^{14}C (maybe it was measured but not shown?), maybe this could help disentangle this effect?

Reply: In ^{14}C analysis, it is a standard procedure to correct the data ($^{14}\text{C}/^{12}\text{C}$ ratios) for changes in the $^{13}\text{C}/^{12}\text{C}$ ratio by normalizing the data to a $\delta^{13}\text{C}$ value of -25 permil (PDB), which is representative of pre-industrial wood. This correction allows to compare samples from different environments (Stuiver and Pollach, 1977). As for paleo records, this correction removes potential biases introduced from changing $\delta^{13}\text{C}$ ratios through time (Stuiver and Pollach, 1977). During ^{14}C measurements using the MICADAS, $^{13}\text{C}/^{12}\text{C}$ is analyzed along with $^{14}\text{C}/^{12}\text{C}$ ratios and the correction is automatically performed by the processing software of the system (BATS; Mollenhauer et al., 2021). All FmC values reported in our manuscript thus are ^{13}C -corrected and any influence from changing C4/C3 plant ratios can be excluded.

To avoid that questions arise regarding the impact of isotopic fractionation in our ^{14}C data we added a few lines to the method section mentioning the ^{13}C correction (458-460).

5) Error propagation: Does the calculation of tau include potential errors for equation 3 and 4? The relationship between lipid age and tau has some error to, and so it may be propagated in this work.

Reply: Yes, the calculated τ_{soil} includes errors from both equations. (Note that in the revised version, these equations are numbered as 4 and 5). Uncertainties from $F^{14}\text{C}_{\text{sample}}$ (AMS measurement uncertainty and errors introduced from the blank correction) and t (deposition age/core chronology) are considered for $F^{14}\text{C}_{\text{initial}}$ (eq 4). From eq. 5 we used the error of the slope (40.1 ± 3.9) which is given in ref 31.

Reviewer #2 (Remarks to the Author):

Review and revise the manuscript

Review conclusion: Acceptance after major revision

The article uses the radiocarbon dating method derived from plant derived lipids, combined with temperature and rainfall reconstructed from sediment cores of the eastern Mediterranean that received terrestrial materials from the Nile River Basin, to investigate the dominant control of temperature on carbon turnover in subtropical and tropical soils over the past 18000 years. The research work is solid, the content is rich, the methods are advanced, and it has good innovation. I am very interested in this part of the content, but it is not easy to read the article. The most important issue is that the main research question and conclusion are not very clear, and the writing logic may need to be reorganized for readers to better understand the content. The second question is whether the definition of carbon in the article is reasonable and accurate, which is very important. At least in my opinion, I don't understand what redefining carbon means? And how to determine, does it refer to the rate or quantity of carbon outflow and inflow, how to calculate it is not clearly stated in the article? Then I still don't understand why the reservoir age shift of leaf wax biomarkers can be used to calculate average soil carbon. This part may require more explanation and clarification. The research is very interesting and meaningful, but there are still some issues that need to be modified and resolved. The specific opinions are as follows:

Dear Reviewer,

thank you very much for your positive review of our manuscript. We are very delighted to read that you find our research very interesting and meaningful and that you think it has good innovation. As we understand from your review, the most important point to you is that our main research goals and conclusions need to be more clearly stated. We address this point in our new version of the manuscript and rephrased parts of the introduction (lines 42-52) and the concluding paragraph (330-343). We also extended a few sentences throughout the manuscript to improve the elaborations about our methods, i.e. the justification why we selected the biomarkers and how we can deduce mean soil carbon turnover times (lines 72-90; 186-195).

Please, find below how we address each of your points. The line numbers refer to the track-changes version of the manuscript.

1. What is the meaning of redefining carbon in Line 30? Can't we directly use organic carbon content? How to determine f , does it refer to the rate or quantity of carbon outflow and inflow, how to calculate it, and what is the unit?

Reply: We are not sure what exactly you mean by "redefining carbon". In line 30 we show the formula commonly used to determine the mean turnover time - or residence time - of carbon in soils. This formula was not developed by us and therefore neither something new is defined here nor is something re-defined. We added a citation to the formula to clarify that this formula is background knowledge that has already existed for long time (ref. 6).

As for your questions regarding the units: Carbon stock is the organic carbon content per unit area and is often given in kgC m^{-2} in both, field experiments and data (see Figure 4e). To determine soil organic content, several methods exist to analyze samples from the field in the laboratory. One example is the dry combustion method using an Elementar Analyzer. Besides analysis in the lab data from remote sensing are used. As mentioned in line 32, influx and carbon efflux are equal in steady states (in equilibrium conditions). So, turnover time of soil carbon can be calculated using either influx or efflux. Therefore, it is common practice to determine the turnover time using the net primary

production (NPP), i.e. the carbon influx. For determining NPP the increase in plant biomass per unit area and time is measured in the field, and using remote sensing or modeling approaches. NPP is often expressed in $\text{kgC m}^{-2} \text{yr}^{-1}$.

We added the units to lines 31 and 32 in order to clarify.

2. Why Lines 36-38 said that the impact of water climate on soil carbon turnover time plays a strong controlling role in low latitude climate, more important than temperature? I don't understand. Is there any theoretical support?

Reply: This statement is based on observations from refs. 4,12,13 which are given in these lines. For example, in ref. 4 (Carvalhais et al., 2014), ecosystem turnover times are correlated against temperature and precipitation to investigate regional differences in the dependence of turnover to these two climate variables. The results suggest that relatively strong relationships with precipitation tend to especially occur in the low to mid latitudes. Carvalhais et al. (2014) concluded that in these regions, precipitation overrides temperature effects as relatively weak correlations with temperature were observed at these places. One reason for this pattern may be that in these regions, temperature variability is small throughout the year while rainfall variability is large throughout. However, the interaction of temperature and precipitation effects on turnover times are still not fully understood. We refrain from elaborating and speculating about potential mechanisms that may explain this observation in the introduction as this would distract from the major focus of our study. By finding that temperature is the major driver of τ_{soil} over the past 18 thousand years the focus is set on the effect of temperature.

3. Lines 68-72: The two biomarkers (e.g., long-chain n-alkanes and n-alkanoic acids) were a bit suddenly proposed here, I suggest to add some descriptions for explaining why you choose these two biomarkers? As I am studying modern processes and mechanisms of these two biomarkers for more than ten years, I totally support that you use these two biomarkers to demonstrate the responses of soil carbon cycle to climate changes, but more explanations are needed here. Please refer some references (e.g., n-alkane for Liu and An, 2020 and Liu et al., 2022; ESR; n-alkanoic acids for Liu et al., 2024; SCES).

Reply: To improve the justification for the selection of the biomarkers, we provide more details in our introduction of the biomarkers in lines 68-95. We added more information about their origin and their application as biomarkers for continental environmental changes as well as about the method for deducing τ_{soil} from CSRA of the n-alkanoic acids. We also provide a new reference (Eglinton and Eglinton, 2008 => ref. 24), which summarizes the application of molecular proxies –including plant-wax lipids (alkanes, alcohols and acids) in paleoclimatology. By deciding to provide this single reference instead of the three you suggested, we are able to stay within the limit of 70 references. Having extended this paragraph, we hope that the proposition of the biomarkers now appears more gradual and coherent.

4. Line 151-153: Why n-alkanoic acids reflect a local signal while n-alkane provide a more catchment-integrating signal? We know these two biomarkers originate the same precursor (i.e., acyl-ACP).

Reply: The precursor molecule of both compounds is acyl-ACP. However, in this case we refer to the geographical origin rather than the biogeochemical one. The reasoning for this conclusion that *n*-alkanes and *n*-alkanoic acids have different source areas is given in detail in Meyer et al. (2024) which is also cited in line 151 (ref. 23). A key finding of this study is that δD of *n*-alkanes and *n*-alkanoic acids (δD_{wax}) differ substantially from each other (Figure 1). The different trends throughout the past 18 kyrs attest to different rainfall regimes and consequently different source areas within the Nile catchment. When comparing the δD_{wax} data from core GeoB7702-3 to δD_{wax} records from Lakes Tana and Victoria (sources of the Nile River) and to data from the Red Sea (Mediterranean realm) it becomes clear that *n*-alkanoic acids reflect rainfall in the Mediterranean winter rainfall zone (Nile delta region) while the *n*-alkanes also reflect the African summer monsoon rains. To illustrate this, we provide Figure 1, which is from Meyer et al. (2024) (Records a-e are of interest). Today, the

Mediterranean winter rains supply a small band along the coast while monsoon rains influence the catchment area south of $\sim 12^\circ N$. The hyperarid desert where vegetation is extremely scarce lies in between. Accordingly, Meyer et al., 2024 conclude that the *n*-alkanoic acids provide a regional signal from the delta (Mediterranean rainfall zone) and that *n*-alkanes provide a more catchment integrating signal. This provenance pattern probably owes to differences in the vulnerability of the two compounds towards degradation. *n*-alkanoic acids are relatively labile compared to *n*-alkanes and probably get more efficiently degraded during transport. Most likely fatty acids from the headwaters did not reach the Mediterranean Sea throughout the last 18 kyrs. As elaborated in Meyer et al. (2024) similar observations regarding provenance of leaf wax biomarkers were previously made in other large river catchments, e.g. the Ganga-Brahmaputra and Congo Rivers (Galy et al., 2011; Hemingway et al., 2016).

As we cite Meyer et al. (2024) in lines 151, we did not change the paragraph. If interested, the reader can easily access the detailed discussion of the provenance patterns following the reference (ref. 23).

Figure 1: (a)-(c) δD_{wax} records from Lakes Tana (Costa et al., 2014), Victoria (Berke et al., 2012) and Tanganyika (Tierney et al., 2008) (d) δD_{wax} *n*-alkanoic acids (orange: *n*-C26:0; green: *n*-C28:0; Meyer et al., 2024) along with δD_{wax} *n*-alkane (blue: *n*-C31; Castañeda et al., 2016) from core GeoB7702-3 and δD_{wax} based on the *n*-C30:0 alkananoic acid from core DSDDP 5017-1, Dead Sea (black; Tierney et al., 2022). (e) Ratio of tetra and penta-methylated brGDGTs from the Nile deep sea fan (Ménot et al., 2020) oxygen isotopic composition of the planktic foraminifera species *Globigerinoides ruber alba* from the Levantine Basin (Revel et al., 2010; 2015). The light grey and dark grey shadings mark the episodes of the AHP and the "Green Sahara", and their optimum. LGM: Last Glacial Maximum; HS1: Heinrich Stadial 1; B/A: Bølling-Allerød interstadial; YD: Younger Dryas

5. Does the topic consider adding marine sediments? The research is not focused on traditional soils.

Reply: Our work is based on a marine sediment core but this fact does not matter for our endeavor. Despite being marine, the core forms a suitable archive for reconstructing continental changes as we choose terrigenous biomarkers for radiocarbon analysis and the reconstruction of τ_{soil} . As the n -C_{26:0}, n -C_{28:0} alkanolic acids and the n -C₃₁ alkane are specific biomarkers for terrestrial higher plants and a bias from marine algal or bacterial biomass can be ruled out.

6. At the beginning, the research site was in the Nile Delta region, and later it moved to tropical and subtropical regions as well as global analysis. I don't know how this changed in between. Can the Nile Delta region represent subtropical regions?

Reply: The delta is the main origin of the n -alkanoic acids as concluded in Meyer et al. (2024) (ref. 23) and accordingly the signals of the n -alkanoic acids reflect environmental change in the Nile delta in the first place. In line 152 we state that the n -alkanes provide catchment integrating signals. We find that the ¹⁴C signals in n -alkanoic acids and n -alkanes are very similar (supplementary figure S2). This observation is the key to justify the extrapolation of the τ_{soil} results to the entire Nile catchment. From the similarity we conclude that changes in soil carbon turnover must have been similar in the entire catchment (line 162-165).

The Nile catchment is vast and extends over the northeastern African tropics and subtropics. So it is directly representative of northeastern Africa. The extrapolation to the global scale is made because CSRA data from (sub-) tropical areas in Asia (Ganga-Brahmaputra catchment; ref. 16) suggest similar magnitudes in change of τ_{soil} (lines 283-286; Table 1 in the manuscript). In view of these similarities in two datasets from the two continents, we consider it likely that these changes in τ_{soil} may have occurred globally across the (sub)- tropics. However, more data from other regions (e.g. South America or Southeast Asia) is necessary to verify this assumption.

We extended the sentence in lines 286-289 to clarify our justification for the extrapolation to the global (sub-) tropics. We also rewrote the introduction of the Nile catchment to stress that it is representative of a relatively large (sub-)tropical area in northeastern Africa to improve the justification of the selection of this study area (lines 56-61).

7. I didn't understand the part about the time in the article. It started in the environmental changes in the Nile Delta region in the previous 18 years, and then it was before glaciers until the New Century. How did it change during this period? What did the environmental changes in the Nile Delta region in the previous 18 years mean? What is the connection to the other content? I don't quite understand?

Reply: Our manuscript is about the last 18 thousand years and discusses changes in τ_{soil} during the last deglaciation. When referring to the time before present we express the time in kiloyears before present (18 kyrs BP). Perhaps the confusion arose because we wrote "18,000 years" in the abstract but "18 kyrs" in the following main text. In the new version of the article we indicate that "18,000 years" is termed "18 kyrs" in the following (line 47).

8. In my opinion, Lines 168-175 has no theoretical support.

Reply: When revising our manuscript, this paragraph was removed.

9. Lines 194-195: the hydrogen isotopic composition of paleo precipitation (δD_p), serve as a common proxy for the amount of rainfall? This needs a detailed explanation, because our modern investigation showed that δD_p is mainly controlled by temperature at the global scale, instead of rainfall amount, only in some intensive monsoon regions, the δD_p is dominantly controlled by rainfall amount, so please supplement some modern data for supporting this inference.

Reply: It is correct that temperature exerts strong control on the δD of precipitation on a global scale. In low latitude regions where temperature variations are often small but variability in rainfall is large – such as monsoon regions – the amount effect is the dominant control on δD of precipitation. This is the reason why dD of n-alkanes and n-alkanoic acids (δD_{wax}) has often been applied as a proxy for rainfall variability in Africa to reconstruct changes in the African monsoon (Castaneda et al., 2016; Berke et al., 2012; Tierney et al., 2008 and many more). The δD_{wax} from our n-alkanoic acids is interpreted as a signal from the Nile delta region (Meyer et al., 2024 => ref. 23) and is outside the area of influence of the monsoonal rains. As for the coastal areas of North Africa that are influenced by Mediterranean winter rainfall, Tierney et al. (2017) as well as Goldsmith et al. (2017; 2019) show that the amount effect exerts dominant control. The δD_{wax} record from the n-alkanoic acids that we use in our current study is intensively discussed in our previous paper that focusses on rainfall variability in the Nile delta region during the last deglaciation (Meyer et al.; 2024 => ref. 23). Below we show a figure from this study showing the δD_{wax} of n-alkanoic acids and n-alkanes together with the TEX_{86} -based SST record. The development of δD_{wax} of the n-alkanoic acids clearly differs from the development of SST which is a strong indication that the amount effect dominated over temperature during the past 18,000 years. Accordingly, we consider the δD_p record (which is deduced from the δD_{wax} by correcting it for changes in the abundance of C3 and C4 plants via the $\delta^{13}C_{wax}$) a robust proxy for rainfall amount. So the use of this record to investigate the link between precipitation and soil-carbon turnover times in our present study is justified.

For the revised version of the manuscript, we extended the paragraph encompassing lines 191-195 to strengthen the selection of δD_p as proxy of rainfall amount in the Nile river catchment.

Figure 2: Figure adopted from Meyer et al. (2024). δD_{wax} from three plant-wax homologues is plotted together with SST reconstructions from the Eastern Mediterranean (data from Castañeda et al. (2010) and summer insolation (Berger and Loutre, 1991). δD_{wax} is corrected for changes in ice-volume.

10. Why do you say that in Lines 277? Is there any data support for non-tropical and subtropical areas? Doesn't this change exist in non-tropical and non-subtropical regions?

Reply: The conclusion in line 277 was made because there are similarities between two datasets from two continents, i.e., the Ganga-Brahmaputra catchment (Asia) and the Nile catchment (Africa). Specifically, it is the order of magnitude of change in τ_{soil} that occurred since the last glacial that is similar at both sites. Data from two river catchments that reflect sub-tropical to tropical areas on two continents make it very likely that the changes in τ_{soil} that we infer for the Nile catchment are not a regional phenomenon. Therefore, we hypothesize that similar changes were common across the subtropics globally. However, more data is needed to confirm this hypothesis (e.g. from South America, western Africa or Southeast Asia).

As for the non-tropical areas, the retreat of permafrost and ice sheets probably had substantial effects on carbon storage, turnover times and release of CO_2 into the atmosphere. The retreat of permafrost from these regions is considered one of the mechanisms fueling the deglacial rise in atmospheric CO_2 and the concurrent decline in atmospheric radiocarbon content. Therefore, many previous studies focused on the northern high latitudes (e.g., Ciais et al., 2013; Winterfeld et al., 2018; Meyer et al., 2019 and more) using radiocarbon data and modeling. By contrast, only very few data exist for the subtropics and tropics (to our best knowledge, Hein et al., 2020 (ref.16), and ours) and very little is known about soil-carbon cycling and the impact on atmospheric CO_2 and $\Delta^{14}C$. Addressing this gap of knowledge is one of our major research goals. We added a small paragraph to the introduction to clarify this (lines 48-52). In our paper we do not address the extra-tropics as they are beyond the scope of our study.

11. Should different vegetation zones in Figure 1 be marked with different colors.

Reply: We decided to leave the figure as is since we intend to highlight the location and extent of the Nile catchment. We think that this focus would get lost, if the vegetation zones were colored. No action taken.

12. It would be better to mark the year on the horizontal axis in Figure 2.

Reply: Thank you for the suggestion. However, no changes were made as it is common practice to plot kyrs on the x-axis in paleo-studies. We feel indicating years instead of kyrs would make the axis labels more difficult to read, as it would require displaying numbers with up to 5 digits.

13. It seems unreasonable to use surface temperature instead of sea surface temperature in the eastern Mediterranean in Figure 3

Reply: Perhaps this is a misunderstanding. We do not use surface air temperatures (SAT) instead of sea surface temperature (SST). In fact, it is the other way around because no SAT records are available. If there were some, we would have used them instead of SST because SAT would be the best parameter to approach soil temperature in the Nile catchment. As explained in the caption of figure 3 in the initial version of the manuscript we think this that SST still is an appropriate record to use. It is very likely that SST and SAT in the Nile delta region developed similarly due to heat exchange between the sea surface and the overlying air. Other work from the Mediterranean realm concludes the same. For example, Bar-Mathews et al. (2003) summarize in their paper that similarities between $\delta^{18}\text{O}$ records from terrestrial (speleothems) and marine (sediments, foraminifera) archives suggest that SST and land temperatures developed similarly during the last deglaciation and the Holocene.

During the revision we shifted the justification for the selection of the SST record from the figure caption to the main text (lines 186-191) as we felt it this information was a bit hidden previously.